# Antioxidant Activity of Aqueous Extracts Obtained from By-Products of Grape, Olive, Tomato, Lemon, Red Pepper and Pomegranate

**DOI:** 10.3390/foods13121802

**Published:** 2024-06-07

**Authors:** María Luisa Timón, Ana Isabel Andrés, María Jesús Petrón

**Affiliations:** Food Technology Department, School of Agricultural Engineering, University of Extremadura, 06007 Badajoz, Spain; aiandres@unex.es (A.I.A.); mjpetron@unex.es (M.J.P.)

**Keywords:** aqueous extract, by-product, polyphenol, vitamin C, carotenoid, in vitro antioxidant activity

## Abstract

The aim of this work was to study the antioxidant potential of aqueous extracts obtained from different by-products. The effectiveness of these extracts was compared with that of rosemary extract. Total phenol carotenoid and vitamin C contents, as well as in vitro antioxidant activity, were assessed. Phenol content was positively correlated with in vitro antioxidant activity in extracts, while carotenoids showed a less clear relationship. Vitamin C was associated with antioxidant activity in lemon and pepper pomace extracts. Extracts from olive, grape, and lemon by-products displayed the highest antioxidant activity (radical scavenging activity), this being similar to the activity of rosemary extracts. Moreover, the phenolic profile of the extracts was analyzed, revealing diverse phenolic compounds. Rosemary extracts contained the highest variety and quantity of phenolic compounds, while olive pomace extracts were rich in hydroxytyrosol and 4-hydroxybenzoic acid. Lemon and pepper extracts contained high amounts of tyrosol, and tomato extracts had abundant epicatechin. The PCA analysis distinguished extracts based on in vitro antioxidant activity, phenol, carotenoid, and vitamin C content, along with their phenolic compound profiles. This study emphasizes the capacity of aqueous extract by-products as valuable sources of antioxidants and highlights the importance of understanding their bioactive components.

## 1. Introduction

Nowadays, consumers are increasingly more concerned about the safety of synthetic additives in foods. Consequently, novel antioxidants from natural sources are receiving attention in the food industry [1], and there have been numerous investigations focused on meeting the increasing demand for natural products by consumers [2]. Food industry by-products are a good source of natural antioxidants and represent an alternative to currently used conventional antioxidants [3].

The food industry generated around 3 million tons of waste in Spain in 2019 [4]. Disposal or incineration should not be promoted, whereas revalorization policies or treatments should be conducted in compliance with the environmental legislation in the European Union, bolstering the transition towards a circular economy [5]. A promising way to enhance the value of food wastes is to transform them into commercially useful products, as they are abundant in bioactive compounds like vitamins, minerals, amino acids, polyphenols, fiber, carotenoids, etc. [6]. In this sense, plant by-products have been described as a natural source of antioxidants [7]. Tocopherols, carotenoids and phenolic compounds from by-products could be considered viable and safe alternatives for human consumption as food additives [8].

The tomato industry is one of the most important sectors in Spain, and it yields huge amounts of by-products, known as tomato pomace [9]. This consists of a mixture of tomato peels, cores, culls, pulp and crushed seeds, and it is a encouraging alternative due to its nutritional properties and antioxidant potential, containing carotenoids, polyphenols, fiber, minerals and vitamins [10].

Olive pomace stands out as the main by-product of the olive processing industry, characterized by its solid composition, comprising skin fragments, pulp, kernel pieces and residual oil. Recognized for its potential, olive pomace could be considered a significant natural reservoir of phenolic compounds, which, following proper refinement, could serve as a valuable food antioxidant [11]. Spain is in the lead as the primary producer and exporter of pomegranates (*Punica granatum*) in Europe [12]. Pomegranate peels, a by-product of juice industries, are notable for their potent array of antioxidants and anti-inflammatory compounds such as polyphenols, vitamin E, sterols, and natural estrogens [12].

Grape pomace stands as the main residue produced by the wine industry, comprising three different components: seeds, stems, and skins. This waste is very interesting due to its rich range of polyphenols, primarily composed of flavonoids, like anthocyanins, flavonols and flavanols such as condensed tannins, which are renowned for their high antioxidant and antimicrobial properties [13]. The citrus genus is the most important fruit tree crop in the world, and lemon is the third most important citrus species with a worldwide production of 7.3 million tons [14], Spain being the first producer of lemon and the second lemon juice producer in the world [15]. Several studies highlighted lemon as an important health-promoting fruit rich in phenolic compounds (mainly flavonoids) as well as vitamins, minerals, dietary fiber, essential oils and carotenoids [16]. Lemon fruit has strong commercial value for the fresh product market and food industry. Moreover, the lemon industry generates high amounts of wastes and by-products, such as non-conforming fruits, pomace, or lemon peel, which constitute an important source of bioactive compounds [17].

The red pepper variety (*Capsicum annum* L.) is highly demanded by the global food industry. After processing, by-products are produced and discarded as non-marketable products, such as peels, stalks, seeds, and unused flesh, that are generated after different stages of the industrial processing [18]. Peppers are a good source of several health-promoting compounds such as vitamin E and C, phenolic compounds, particularly flavonoids, and carotenoids [19].

The extraction of bioactive compounds involves separating them from plant tissues by using different extraction procedures. Numerous studies have focused on the development of techniques for the recovery of these bioactive compounds. Different extraction techniques yield highly varied results in terms of performance, extraction specificity, time, cost, and environmental impact [20]. These latter aspects are of particular importance and will determine the reliability and quality of subsequent determination. In this regard, there is a real need to choose extraction methods for these compounds that are economical and environmentally friendly, reduce the extraction duration, and enhance yields of bioactive compounds, all while preserving their biological efficacy [20].

Green solvents for revaluing by-products from food industry have received much attention recently [21]. In this sense, the use of water to obtain bioactive extracts from by-products can be considered an alternative to the use of conventional organic solvents. Water has been suitable for extracting compounds such as phenols, vitamin C, saponins and terpenes from different vegetable by-products [20]. Interestingly, despite the poor solubility of carotenoids in water, Mussagy et al. [22] were able to recover some carotenoids from *Rhodotorula glutinis* using this solvent. To the best of our knowledge, there are no studies focused on the extraction of phenol compounds from grape, olive, tomato, lemon, red pepper, and pomegranate by-products using water as a solvent. The antioxidant activity of the mentioned extracts should be compared with that of a positive control group. Rosemary extract is used for the purposes of preventing oxidation in food, creating food additive E-392, and representing an alternative to synthetic antioxidants. Several investigations have concluded that rosemary antioxidant activity is molecularly superior to, or at least equivalent to, that of synthetic antioxidants commonly used in the food industry [23].

The aim of this work was to obtain aqueous extracts with in vitro antioxidant activity from tomato (TOM), grape (GRA), olive (OLI), pomegranate (POM), lemon (LEM) and red pepper (PEP) by-products. The effectiveness of these extracts compared with that of rosemary (ROS) extract, a commercialized extract with proved antioxidant activity, was also studied.

## 2. Materials and Methods

### 2.1. Chemical Standards and Reagents

The reagents sodium carbonate, di-sodium hydrogen phosphate, sodium carbonate, potassium ferricyanide, ethylenediamine tetra acetic acid (EDTA) and gallic acid were purchased from Scharlau (Barcelona, Spain). Folin–Ciocalteu’s phenol reagent, absolute ethanol and trichloroacetic acid (TCA) were purchased from Panreac (Castellar del Vallès, Barcelona, Spain). 2,2-diphenyl-1-picrylhydrazyl (DPPH) free radical, Iron (II) Chloride 4-hydrate, and (L)-Dehydroascorbic acid were purchased from Sigma Chemical Co. (Steinheim, Germany, and St. Louis, MO, USA). The following standards were used: lycopene, β-carotene, naringenin, rutin, caffeic acid, caftaric acid, catechin, neochlorogenic acid, cumaric acid, ellagic acid, epicatechin, hydroxybencene, hydroxyphenilethanol, hydroxytirosol, oleuropein, punicalagin, quercetin, syringic acid, vanillin, and verbascoside (Merck KGaA, Darmstadt, Germany).

### 2.2. Extract Preparation

Agroindustrial by-products were sourced from CICYTEX-La Orden (Guadajira, Spain). The proximate composition is depicted in Table 1. Moisture, ash and protein analyses were performed in accordance with the AOAC methodology [24], and fat content was assessed using the Folch method [25], while fiber was assessed according to the AOAC methodology [24].

Rosemary extract, used as a positive control, was supplied by IFF Murcia Natural Ingredients, S.L.U. (Spain). Extracts were prepared following the protocol by Andrés et al. [26], as follows: By-product samples were placed in trays, spread in 2 cm thick layers and dried at 112 °C until constant and similar moisture contents (6%) were achieved. Samples were then ground in water (1:10 *w*/*v*) and acidified to reach a pH of 2.5–3 by using acetic acid to improve extraction and stabilize phenol compounds. Homogenization was carried out at 30 °C for 2 h with continuous stirring. Once extraction was finished, samples were cooled, filtrated through a 45 μm diameter pore filter and centrifuged at 3000 rpm for 10 min. The supernatant was recovered, lyophilized, and stored for further use. Liquid extracts were prepared (1 mg/mL) for antioxidant activity and phenol content analysis.

### 2.3. Radical Scavenging Activity (RSA) Assay

The antioxidant activity of the extracts, based on the scavenging activity of the stable 1,1-diphenyl-2-picrylhydrazyl (DPPH) free radical, was determined using the method described by Broncano et al. [27]. The ability to scavenge the DPPH radical was expressed as the inhibition percentage and was calculated using the following formula:
RSA (%) = [(A_control_ − A_sample_)/A_control_] × 100where A_control_ is the absorbance of the control (DPPH solution without sample), and A_sample_ is the absorbance of the extract (the DPPH solution plus the test sample). The same concentration of ascorbic acid (vitamin C) was used as a positive control.

### 2.4. Metal Chelating Activity (MQA) Assay

The Fe^2+^ chelating ability of the extracts was estimated using the method by Broncano et al. [27]. The percentage of inhibition of ferrozine–Fe^2+^ complex formation was calculated using the following equation:MQA (%) = [(A_control_ − A_sample_)/Acontrol] × 100where A_control_ is the absorbance of the control (without sample), and A_sample_ is the absorbance of the test sample (FeCl_2_ and ferrozine solutions plus extract). The same concentration of EDTA was used as a positive control.

### 2.5. Reducing Power (RP) Assay

The RP was determined according to the method described by Broncano et al. [27]. Absorbance was measured at 700 nm. Ascorbic acid (vitamin C) was used as a positive control. Results were expressed as mg ascorbic acid per g extract sample.

### 2.6. Total Phenolic Content (TFC)

The total phenolic content in extracts was determined according to the Folin–Ciocalteu procedure [28]. Results are expressed as mg gallic acid equivalents (GAE) per g extract sample.

### 2.7. Vitamin C Content (Vit C)

An amount of 10 mg of dry extract was homogenized with 10 mL of 0.3 mol/L metaphosphoric acid. The solution was filtered through a 0.45 μm membrane filter. Vitamin determination was analyzed as previously described by Rizzolo et al. [29]. Then, 25 μL of the solution was injected into the HP1100 liquid chromatograph on an Inertsil ODS-3 column (5.0 μm particle size, 4.6 mm i.d. × 250 mm), coupled to a photodiode-array detector operating at 254 nm and 0.002 AUFS. The mobile phase was 0.02 M ortho-phosphoric acid, and the flow rate was adjusted to 0.7 mL/min. The column was operated at room temperature (25 °C), and chromatographic peak data were integrated up to 30 min. Concentrations of vitamin C were calculated from the integrated areas of the samples and a calibration curve of the standard.

### 2.8. Licopene and β-Carotene Analysis (TCC)

An amount of 20 μL of extract (20 mg/mL) was filtered through a 0.45 μm membrane filter and analyzed via HPLC [30]. Identification of β-carotene and licopene was achieved by comparing the retention time and UV-visible absorption spectrum with those of the standard (Sigma-Aldrich, Steinheim, Germany), and quantification was achieved using a calibration curve of the standards. Carotenoid content was expressed as mg of carotenoids (lycopene+ β-carotene) per g extract.

### 2.9. Phenolic Compounds by HPLC

Phenolic compounds were analyzed using a HP1100 liquid chromatograph equipped with a diode array detector and an Inertsil ODS-3 column (5.0 μm particle size, 4.6 mm id, 250 mm), preceded by the use of an Inertsil ODS-3 guard column (5.0 μm, 4.0 mm × 10 mm), as previously described by Timón et al. [31]. HPLC separation was performed using a mobile phase consisting of 2.5% formic acid in water (phase A) and 2.5% formic acid in acetonitrile (phase B), with a flow rate of 1 mL/min used under the following gradient: 0 min, 5% B; 13 min, 11% B; 16 min, 13% B; 20 min, 14% B; 22 min, 15% B; 25 min, 20% B; 28 min, 25% B; 30 min, 30% B; 40 min, 5% B. The runtime was 40 min. The injection volume was 25 µL, and analytes were detected at a wavelength of 280 nm.

The identification of phenolic compounds was performed by comparing the retention times with those of standards, and quantification was carried out with calibration curves.

### 2.10. Statistical Analysis

Statistical analysis of the data was conducted using one-way ANOVA, followed by a comparison of means between different groups using the Tukey test. Correlations among the collected data were assessed using Pearson’s correlation coefficient (r). Additionally, principal components analysis (PCA) was employed to explore the relationships among the antioxidant activity values, total phenol content, and individual phenolic compounds present in the samples. All these analytical procedures were carried out utilizing SPSS software package version 21.0.

## 3. Results and Discussion

### 3.1. In Vitro Antioxidant Activity and Bioactive Components of the Extracts

Correlations between in vitro antioxidant activity, total phenols, total carotenoids, and vitamin C content of the extracts are shown in Table 2. In vitro antioxidant activity in vegetable and fruits has previously been related to high phenol content [32]. However, the relationship between carotenoids and this activity is not so evident, although, in general, a positive correlation has been found [30]. In this sense, Hanson et al. [33] explained that lycopene and β-carotene content were related to antioxidant activity in tomatoes, together with the phenol and ascorbic acid contents, and Sora et al. [34] attributed this antioxidant activity to capsaicins and carotenoids in pepper. Additionally, other researchers have stated that vitamin C exhibits potent antioxidant properties [35]. In this regard, antioxidant activity has been related to high vitamin C content in citrus fruits [36] and pepper [37]. All of these findings agree with the results obtained in this study, where the antioxidant activity, expressed as RSA, was correlated with TFC and vit C (r = 0.496, 0.623, respectively; *p* < 0.05), and TFC was also positively correlated with MQA and RP (r = 0.747, 0.591, respectively; *p* < 0.01). However, TCC was not correlated with any of the in vitro antioxidant parameters.

Scientific research that has concentrated on the chemical characterization and in vitro antioxidant potential of extracts derived from vegetable by-products has been widely published in recent years, including lemon, red pepper, pomegranate, tomato, grape and olive pomace [3,38]. Nevertheless, comparing results is challenging both internally and in comparison with previously published data, primarily due to the variations in extraction solvents and units utilized for data expression. For this reason, in the present experiment, rosemary (*Rosmarinus officinalis* L.) extract was used as a positive control extract due to its well-documented antioxidant activity [23]. Results of the in vitro antioxidant activity, total phenol, total carotenoid, and vitamin C contents of the extracts are shown in Table 3. As can be observed, values of RSA show that olive (95%), grape (95%) and lemon (93%) pomaces were able to inhibit in vitro oxidation more efficiently than the rest of extracts (tomato, pomegranate, and pepper pomaces) (*p* < 0.01), with no significant differences among these three extracts and the one used as a positive control (rosemary, 99%). These results are consistent with those of previous studies on rosemary, grape and olive pomace, where these extracts showed higher RSA values than other vegetable extracts [39,40,41]. However, other authors even reported a higher RSA in lemon than in rosemary extracts [42]. In this mentioned study, extracts were obtained using ethanol, and hence, antioxidant activity could be associated with compounds different than the ones present in aqueous extracts of this experiment. In addition, according to Herrera-Pool et al. [43], the type of solvent can influence the antioxidant activity of the extracts. Overall, extracts obtained with more polar solvents showed higher antioxidant activity, as the polar phase of the extract contributes to the inhibition of radicals [44]. Nevertheless, values of RSA for aqueous grape extracts of this experiment were higher than those obtained using water and ethanol (30:70) in the study by Zhu et al. [45], and values of aqueous lemon extracts were also higher than those obtained when different solvents were used in Al-Qassabi et al. [46]. In the case of aqueous tomato extracts, the RSA percentage was much lower than that obtained by Elbadrawy and Sello using an organic solvent [47].

Results for MQA also showed the highest antioxidant activity for olive (59%) and grape (61%) pomaces (*p* < 0.01). Olive, grape and pomegranate (49%) pomaces showed higher MQA than rosemary (39%), whereas tomato (27%), lemon (19%) and pepper (6%) pomaces showed antioxidant activity below that in the control extract (*p* < 0.01). In a previous study, Andrés et al. [41] also described observing higher MQA in extracts from olive and grape pomaces than in tomato and pomegranate ones. However, Santana-Méridas et al. [48] found higher values of this parameter in rosemary than in grape extracts, these extracts being obtained using ethanol as a solvent.

A similar tendency was found for the RP values, with olive and grape pomaces showing the highest antioxidant activity (*p* < 0.01). Extracts from olive pomace were the most efficient at inhibiting oxidation (1908 mg/g), followed by grape pomace (1080 mg/g) and rosemary (584 mg/g) (*p* < 0.01). On the other hand, lemon (101 mg/g) and pepper (41 mg/g) presented the lowest RP values (*p* < 0.01). Santana-Méridas et al. [48] also reported higher RP values in grape extracts than rosemary ones, although these authors used ethanol for extraction. The highest RP being observed in extracts from olive and grape pomaces compared with those from tomato and pomegranate has also been described by Andrés et al. [41]. Morales-Soto et al. [49] found the greatest values for this parameter for grape and lemon extracts followed by those from pepper, pomegranate and tomato. In their study, extracts were obtained using methanol/water (80:20) as solvent.

Given these results, it appears that RSA has direct relationship with the other activities measured (MQA and RP). In this sense, Pearson’s correlation coefficients showed that RSA was positively related to MQA and RP (r = 0.469, 0.541, respectively; *p* < 0.05), and MQA and RP were also correlated (r = 0.709, *p* < 0.01), indicating that compounds present in the extracts (phenols, vitamin C and carotenoids) could display various in vitro antioxidant activities. Specifically, TFC was positively related to RSA (r = 0.496, *p* < 0.05), MQA and RP (r = 0.747, 0.591, respectively; *p* < 0.01), and vit C was positively related to RSA (r = 0.623, *p* < 0.05). Other authors have also described different antioxidant mechanisms in phenol extracts from spices [50] and have related vitamin C to radical scavenging mechanisms [51]. On the other hand, no mechanism has been related to carotenoids from aqueous extracts of this study, despite both radical scavenging and reducing power mechanisms having been described for carotenoids in orange by-products [52].

Values of TPC in by-product extracts were quite high in this experiment compared with those found in other studies using different technologies [53]. Rosemary extract (960 mg/g) obtained the highest quantities followed by olive (884 mg/g) and pomegranate samples (809 mg/g). However, tomato extracts (138 mg/g) presented the lowest values (*p* < 0.01). Numerous studies have highlighted that rosemary has a high concentration of bioactive components, mainly phenolic compounds, which agrees with our results [54]. Moreover, Andrés et al. [41] also described the highest content of phenols in pomegranate and olive pomaces.

Several studies have indicated that, typically, higher phenolic content tends to correlate with stronger antioxidant activity [32,55]. Therefore, it could be hypothesized that the higher content in phenols led to the greater RSA values in rosemary and olive pomace extracts, as well as the higher MQA and RP in the latter extract. However, the high values of RSA (95%) and MQA (61%) in grape pomace extracts do not seem to be due to higher TFC values (320.23 mg/g). In this regard, Lingua et al. [56] found that antioxidant activity in wine was not related to the total phenolic content but rather to the phenolic profile, as not all phenolic compounds use the same mechanisms to delay oxidation, and some compounds may be more efficient than others [57]. For instance, flavonoids with a large number of hydroxyl groups are more efficient than compounds with a lower degree of hydroxylation [57]. Castro-López et al. [58] also indicated that antioxidant activity cannot be solely attributed to the content of phenolic compounds but also to the action of different antioxidant compounds and their interaction. On the other hand, the highest content of vitamin C in lemon pomace extracts (3040.99 µg/g) (*p* ≤ 0.001) could be responsible for the high RSA values in this sample. Other authors have proposed that vitamin C is a powerful antioxidant and could enhance the antioxidant potential of the extracts [35]. Generally, the antioxidant activity of citrus fruits has been associated with their high vitamin C content [36]. Vitamin C values in pepper pomace were also high (1390 µg/g) in this research, which is consistent with results found in other studies [59]. However, in vitro antioxidant activity in pepper pomace was not so relevant in comparison with rosemary and olive extracts. In this regard, Zhuang et al. [37] described that the antioxidant capacity of red pepper is mostly linked to ascorbic acid rather than the content of phenolic compounds, and this activity may also be attributed to other metabolites such as capsaicins or carotenoids. In this study, the in vitro antioxidant activity of pepper pomace extract was also not significantly affected by the carotenoid content, despite the high value of TCC (647 µg/g) in these samples. Similar results were found for tomato pomace extract, which presented the highest values of TCC (1325 µg/g) (*p* ≤ 0.01) but did not lead to high antioxidant activity. Hanson et al. [32] explained that the content of lycopene and β-carotene was related to antioxidant activity in tomatoes of different varieties. This discrepancy could be justified by the observations by Mussagy et al. [60], who indicated that carotenoids are vulnerable to certain instability effects depending on the solvent used. Aqueous extraction could affect the stability of these compounds, affecting their antioxidant potential.

### 3.2. Phenolic Profile of Extracts

Antioxidant activity cannot be solely attributed to the content of phenolic compounds but also to the action of different antioxidant compounds and their interaction with each other [58]. Studying the phenolic profile of by-product extracts provides a more comprehensive understanding of their activities, synergistic interactions, structures and physicochemical properties, specifically the solubility of these compounds. Phenolic compounds include one or more hydroxyl groups directly attached to an aromatic ring, which partially determine their solubility in different solvents [61]. Water has been suitable for extracting polar compounds, such as phenols from potato peel, apple waste, grape pomace, blackberry, almond and asparagus by-products, as well as brewer’s spent grain [31,62,63,64,65,66].

Table 4 illustrates the phenolic compound content identified via HPLC in the various aqueous by-product extracts. The standards selected for this study are those phenolic compounds generally identified in fruits and vegetables [67,68]. Chemical structure of these compounds is shown in Appendix A (https://pubchem.ncbi.nlm.nih.gov). As was expected, most of the phenolic compounds were identified in higher quantities in rosemary extracts. Interestingly, the number of compounds and their content were low in olive pomace extracts, except for 4-hydroxybenzoic acid and hydroxytyrosol, which showed the highest content in these samples (*p* < 0.001). Therefore, it could be suggested that the higher antioxidant activity in olive pomace extracts (Table 3) may be attributed to these compounds. In this context, many authors have highlighted the high antioxidant properties of both compounds [69,70].

As can be observed, tyrosol (2,4-hydroxyphenylethanol) was one of the phenolic compounds extracted in a greater quantity from all samples, this compound being more abundant in rosemary and in lemon and pepper by-product extracts (*p* ≤ 0.001). Tyrosol has been found to be abundant in samples of olive oil, wine, and other plants [61], but, as far as we know, it has not been previously identified in rosemary or lemon by-products. However, Lobato-Ureche et al. [71] found this compound in pepper plants. Curiously, it was not identified in olive pomace extracts in the present experiment. In this regard, Loschi et al. [11] indicated that this compound is not abundant in olive pomace, whereas hydroxytyrosol is more abundant in this by-product. In fact, olive pomace extract presented the highest content of hydroxytyrosol in this study (*p* < 0.001). Tyrosol has many advantageous properties, such as antioxidant, antimicrobial and anti-inflammatory activities [72]. On the other hand, tyrosol has been described as water-soluble and has showed the highest solubility compared with other phenols such as ellagic, protocatechuic, syringic, and coumaric acids [73].

Oleuropein was also present in large quantities in the extracts. Oleuropein is included in a specific group of compounds similar to coumarin, secoiridoids, and is a glycoside formed between elenolic acid and 3,4-dihydroxyphenylethanol (hydroxytyrosol) by an ester bond, as well as a glucose molecule formed by a glycosidic bond. Rosemary extract presented the highest content of this compound (*p* ≤ 0.001). Other authors have also described the presence of oleuropein in rosemary [74]. It is abundant in olive leaves, but its presence has also been described in argan oil [75]. The presence of OH groups in its molecules makes it water-soluble, as demonstrated by Martínez Navarro et al. [76]. Many authors have indicated that oleuropein has potent antioxidant and anti-inflammatory effects [77].

Rutin has been identified in all extracts, this being found in higher quantities in rosemary extracts (*p* ≤ 0.001) than in the rest of samples. It is a flavonoid widely distributed in plants and fruits, is found in over 70 plant species [78], and exhibits many therapeutic properties primarily attributed to its antioxidant potential and anti-inflammatory activity. However, Frutos et al. [79] indicated that its beneficial effects could be influenced by the quantity of it, as well as its bioavailability, which could be hindered by its low solubility in water. In this regard, rutin barely dissolves in ethanol and is highly soluble in methanol [80].

4-hydroxybenzoic acid has been identified in all extracts. This compound has been found in millet, tomatoes, eggplants, potato peel, cherry, plum, raspberry, currant, anise, white mustard, corn, bay leaf, fennel, pecans, vanilla, and coconut [69]. The highest amounts of this compound were found in olive pomace extracts (*p* ≤ 0.001). Other authors have found this compound in samples of virgin olive oil [81]. This phenolic acid readily dissolves in water since, as described by Joshi et al. [69], the acid molecules form hydrogen bonds with available water molecules and, therefore, remain as monomers in the aqueous phase.

Quercetin has been found in fruits, vegetables, and nuts, and exhibits various activities including antioxidant, anti-inflammatory, and antimicrobial activity. Like other flavonoids, it is barely soluble in water [82]. Rosemary extracts showed the highest amounts of this compound (*p* < 0.001). In this regard, Vallverdú-Queralt et al. [83] found quercetin in various aromatic herbs used as spices, such as rosemary, thyme, oregano, cinnamon, bay leaf, and cumin.

Punicalagin is the most abundant component in pomegranate peel with a high molecular weight and is considered a characteristic compound of this part of the fruit. It is a water-soluble ellagitannin, naturally occurring in two anomeric structures, α and β. It is classified in the chemical class of hydrolysable tannins. Various potential therapeutic applications of punicalagin have been proposed, mainly due to its high antioxidant capacity, attributed to the presence of sixteen phenolic hydroxyl groups in its structure [84]. In this study, this compound was found in higher quantities in rosemary extracts (*p* ≤ 0.001), with pomegranate waste extracts showing the lowest contents. After an exhaustive literature review, the presence of this compound in rosemary was not described.

Verbascoside is a water-soluble phenylethanoid glycoside found in several medicinal plants, especially those belonging to the laminar order, which includes oregano, olive, rosemary, thyme, verbena, and acanthus, among others, isolated in different parts of the plant in more than 200 species [85]. Like most of the identified phenols, rosemary extracts showed the highest contents in this phenol (*p* ≤ 0.001).

Caffeic, caftaric, ellagic, syringic, vanillic and coumaric acids were found in higher quantities in rosemary extracts (*p* ≤ 0.001). Lemon pomace extracts also presented the highest contents of caftaric and coumaric acids (*p* ≤ 0.001), and grape pomace presented the highest content of vanillic acid. Tzima et al. [86] also identified caffeic, syringic, and coumaric acids in rosemary. Ellagic acid has been found in fruits such as raspberry, pomegranate, blueberries, strawberries, and nuts [87]. Caftaric acid is the most common hydroxycinnamate in grapes, composed of caffeic acid combined with tartaric acid. Due to yeast metabolism occurring during young wine fermentation, it is believed that caffeic acid is released from caftaric acid [88]. Vanillic acid is a phenolic derivative from eatable plants and fruits and is the oxidized form of vanillin [89]. All these phenolic acids are soluble in water. In this regard, Queimada et al. [73] indicated that the number of hydroxyl groups in the molecule will influence the hydrogen bonds with water, thus affecting the solubilization process.

The highest quantities of catechin were found in rosemary (*p* ≤ 0.01), and the highest quantities of epicatechin were found in lemon and tomato pomace extracts (*p* ≤ 0.001). The group of catechins, belonging to the flavonoid family, contains multiple hydroxyl groups in their molecules, imparting strong antioxidant properties [90]. The main sources of catechin are grapes, wine, apples, pears, berries, coffee, green and black tea [9]. Catechins exhibit low solubility in water [90]. Nevertheless, other authors pointed out that these compounds also exhibit hydrophilic characteristics due to the –OH groups in their molecules [90].The PCA analysis resulted in a two-component model that explained up to 70% of the total variance (Figure 1). The first principal component, PC1 (*x*-axis), accounted for 48% of the data variance, whereas the second principal component, PC2 (*y*-axis), accounted for 22% of the data variance. Extracts from by-products could be visibly separated via PCA analysis based on their in vitro antioxidant activity, total phenol, carotenoid and vitamin C content, and phenolic compound profile. It was observed that rosemary extracts were associated with most of the phenolic compounds (syringic acid, verbascoside, quercetin, rutin, oleuropein, punicalagin, caffeic acid, catechin, cumaric acid and ellagic acid) and radical scavenging activity (RSA). Lemon by-product extracts were associated with vitamin C, besides epicatechin. Tomato and pepper pomace extracts were related, in the left negative quadrant, with the total content of carotenoids (TCCs). Grape and olive pomace extracts were in the same quadrant as TFC, MQA and RP, as well as hydroxytyrosol. Finally, the first principal component separated pomegranate pomace extracts on the basis of TFC.

## 4. Conclusions

The effectiveness of water as a solvent to extract antioxidant compounds from different by-products can be stated. Using an economical and environmentally friendly procedure to extract compounds from underexploited materials presents a viable alternative for the recovery of antioxidant compounds. The comparison with rosemary extract proves that the extraction could be improved by combining it with other techniques. Hence, there is a pressing need to investigate potential methods to explore this environmentally friendly solvent for achieving highly effective extraction of antioxidant compounds.

In general, the higher phenolic, vitamin C, and carotenoid contents in the extracts imply more intense antioxidant activity. However, carotenoids could be vulnerable to certain instabilities due to the use of water, thereby affecting their antioxidant potential. Furthermore, antioxidant activity is also related to the phenolic profile, as not all phenolic compounds utilize the same mechanisms to delay oxidation, and some compounds may be more effective than others.

The phenolic profile of the extracts varied widely, with different phenolic compounds identified in each sample. While some phenolic compounds were consistently present across all extracts, their concentrations varied, contributing to differences in antioxidant activity. This underscores the complexity of antioxidant mechanisms and the importance of considering the synergistic interactions among various phenolic compounds in determining overall antioxidant capacity. In this sense, principal component analysis (PCA) revealed associations between specific phenolic compounds and the antioxidant activity complex.

## Figures and Tables

**Figure 1 foods-13-01802-f001:**
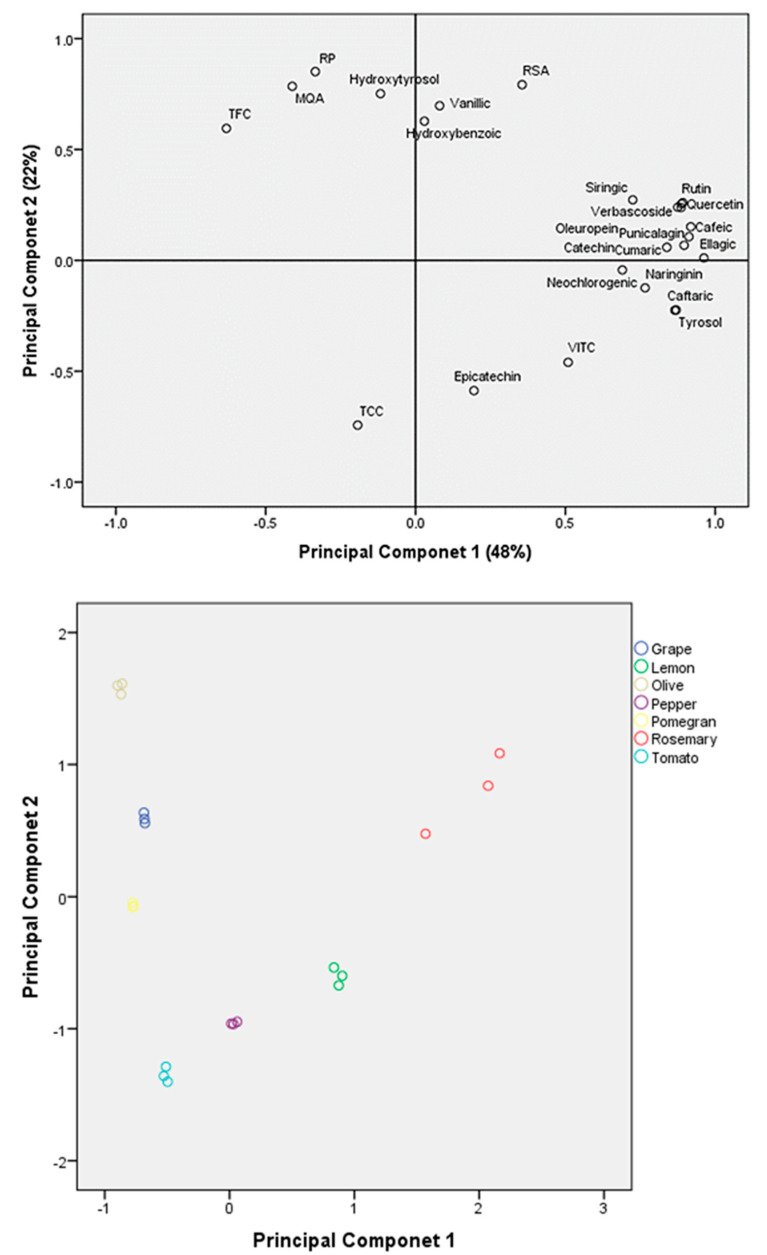
PCA plot of the aqueous extracts of different by-products and variables. For variable names, see tables.

**Table 1 foods-13-01802-t001:** Chemical composition of by-products expressed as a percentage (%). The results shown are the means ± standard error of the mean *(n* = 5).

	Moisture	Protein	Fat	TDF *	Ash
PEP	71.24 ± 3.13	2.82 ± 0.11	8.50 ± 1.39	20.00 ± 2.26	6.90 ± 0.22
LEM	43.28 ± 3.15	5.60 ± 0.15	4.98 ± 1.23	21.30 ± 1.56	6.26 ± 0.41
OLI	45.48 ± 1.73	12.60 ± 0.45	12.58 ± 0.85	38.10 ± 1.89	7.00 ± 0.98
GRA	59.81 ± 2.74	14.00 ± 0.79	11.74 ± 1.62	25.77 ± 2.91	8.30 ± 0.72
TOM	80.03 ± 4.30	14.05 ± 0.77	19.30 ± 1.19	41.63 ± 2.95	5.05 ± 0.40
POM	80.15 ± 2.14	11.10 ± 1.87	10.09 ± 0.47	43.45 ± 1.87	2.80 ± 0.99

* Total dietary fiber. Extracts: PEP: red pepper pomace; LEM: lemon pomace; OLI: olive pomace; GRA: grape pomace; TOM: tomato pomace; POM: pomegranate pomace.

**Table 2 foods-13-01802-t002:** Correlations between the parameters studied using the Pearson correlation coefficient (r).

	RSA	MQA	RP	TFC	TCC	Vit C
RSA	1	0.469 *	0.541 *	0.496 *	0.321	0.623 *
MQA	0.469 *	1	0.709 **	0.747 **	−0.354	−0.406
RP	0.541 *	0.709 **	1	0.591 **	0.128	−0.337
TFC	0.496 *	0.747 **	0.591 **	1	−0.356	−0.438 *
TCC	0.321	0.354	0.128	−0.356	1	−0.234
Vit C	0.623 *	−0.406	−0.337	−0.438 *	−0.234	1

Significant levels: *: *p* < 0.05; **: *p* < 0.01.

**Table 3 foods-13-01802-t003:** RSA (%), MQA (%), RP (mg AAE/g extract), vit C(µg AA/g extract), TFC (mg GAE/g extract) and TCC (μg carotenoids/g extract) of by-products and rosemary (positive control) extracts. The results shown are the means ± standard error of the mean (n = 5).

	RSA	MQA	RP	Vit C	TFC	TCC
PEP	68.95 ± 0.91 ^d^	6.03 ± 0.63 ^f^	41.33 ± 0.46 ^g^	1390.00 ± 8.57 ^b^	240.55 ± 8.36 ^d^	647.23 ± 8.70 ^b^
LEM	93.34 ± 0.01 ^ab^	18.87 ± 0.07 ^e^	101.66 ± 3.95 ^f^	3040.99 ± 4.59 ^a^	391.29 ± 4.64 ^c^	32.5 ± 4.86 ^c^
OLI	95.40 ± 3.63 ^ab^	59.43 ± 3.45 ^a^	1908.45 ± 17.76 ^a^	55.26 ± 6.95 ^de^	884.89 ± 17.80 ^a^	18.64 ± 2.06 ^d^
GRA	94.89 ± 0.18 ^ab^	60.67 ± 4.04 ^a^	1079.72 ± 5.92 ^b^	68.45 ± 14.59 ^de^	320.23 ± 1.51 ^c^	6.96 ± 4.33 ^d^
TOM	48.30 ± 5.81 ^e^	27.47 ± 2.60 ^d^	138.48 ± 3.95 ^e^	463.25 ± 15.65 ^c^	138.84 ± 9.82 ^e^	1325.02 ± 7.78 ^a^
POM	77.99 ± 1.23 ^c^	49.27 ± 1.63 ^b^	205.60 ± 0.00 ^d^	70.29 ± 2.48 ^d^	809.05 ± 3.29 ^b^	122.87 ± 6.78 ^bc^
ROS	99.06 ± 0.49 ^a^	38.51 ± 0.64 ^c^	548.01 ± 4.56 ^c^	624.56 ± 61.44 ^c^	960.6 ± 12.42 ^a^	72.34 ± 6.13 ^c^
P	**	**	**	***	**	**

^a–g^: Values with different letters are significantly different; significance levels: **: *p* < 0.01, ***: *p* < 0.001.

**Table 4 foods-13-01802-t004:** Content in phenolic compounds identified via HPLC in extracts from by-products and rosemary, expressed as µg/g extract. The results shown are the means ± standard error of the mean (n = 3).

	OLI	POM	TOM	GRA	LEM	PEP	ROS	*p*
**Naringenin**	nd	nd	74.18 ± 6.99	nd	22.91 ± 1.34	38.04 ± 7.53	137.37 ± 4.47	ns
**Rutin**	341.12 ± 75.39 ^c^	88.07 ± 3.39 ^d^	664.28 ± 3.37 ^c^	67.97 ± 1.69 ^d^	1748.84 ± 22.39 ^b^	1067.87 ± 29.83 ^b^	10,181.53 ± 1189.50 ^a^	***
**Caffeic acid**	126.53 ± 1.14 ^c^	40.88 ± 0.93 ^d^	104.97 ± 1.35 ^c^	116.51 ± 3.26 ^c^	1082.66 ± 20.25 ^b^	nd	1337.11 ± 21.49 ^a^	***
**Caftaric acid**	nd	75.42 ± 1.60 ^bc^	111.89 ± 0.58 ^b^	62.43 ± 1.27 ^c^	1377.68 ± 8.99 ^a^	1385.58 ± 11.82 ^a^	1568.09 ± 128.49 ^a^	***
**Catechin**	nd	45.35 ± 2.10 ^c^	33.62 ± 3.31 ^c^	79.20 ± 12.96 ^c^	995.44 ± 47.65 ^b^	843.00 ± 61.52 ^b^	2685.02 ± 119.38 ^a^	**
**Neochlorogenic acid**	nd	nd	nd	180.69 ± 11.09	3575.25 ± 256.12	nd	2869.28 ± 743.65	ns
**Coumaric acid**	81.59 ± 4.53 ^c^	74.88 ± 0.10 ^c^	120.42 ± 9.82 ^bc^	80.95 ± 1.88 ^c^	1486.60 ± 1.12 ^a^	nd	1489.66 ± 21.46 ^a^	***
**Ellagic acid**	nd	123.49 ± 12.14 ^c^	199.68 ± 1.23 ^c^	112.66 ± 0.62 ^c^	2455.01 ± 37.54 ^b^	2432.73 ± 4.10 ^b^	5243.48 ± 105.80 ^a^	***
**Epicatechin**	nd	42.85 ± 14.21 ^c^	434.60 ± 14.41 ^a^	27.93 ± 1.37 ^c^	277.88 ± 60.42 ^ab^	nd	196.73 ± 48.40 ^b^	***
**4-Hydroxybenzoic acid**	1559.95 ± 32.62 ^a^	6.64 ± 2.75 ^e^	51.40 ± 0.50 ^d^	79.76 ± 1.62 ^d^	376.70 ± 2.81 ^c^	372.58 ± 38.39 ^c^	1003.12 ± 11.23 ^b^	***
**2,4-Hydroxyphenilethanol**	nd	976.08 ± 6.59 ^b^	1081.76 ± 2.52 ^b^	1144.95 ± 3.19 ^b^	20,194.33 ± 127.35 ^a^	19,549.47 ± 67.61 ^a^	20,992.62 ± 375.84 ^a^	***
**Hydroxytyrosol**	18.65 ± 0.01 ^a^	nd	nd	0.05 ± 0.00 ^c^	1.43 ± 0.15 ^c^	nd	5.18 ± 0.09 ^b^	***
**Oleuropein**	578.68 ± 4.39 ^c^	319.04 ± 18.10 ^c^	nd	372.69 ± 0.77 ^c^	9478.17 ± 27.70 ^b^	7850.84 ± 353.28 ^b^	53,621.42 ± 3642.57 ^a^	***
**Punicalagin**	302.16 ± 4.23 ^c^	266.17 ± 1.86 ^d^	308.00 ± 0.70 ^c^	315.42 ± 2.56 ^c^	5511.87 ± 11.73 ^b^	nd	6295.36 ± 576.41 ^a^	***
**Quercetin**	nd	133.91 ± 23.92 ^d^	58.05 ± 5.41 ^e^	289.44 ± 5.71 ^c^	1559.57 ± 249.90 ^b^	1213.25 ± 14.14 ^b^	9683.01 ± 236.06 ^a^	***
**Syringic acid**	29.54 ± 7.21 ^c^	47.34 ± 3.98 ^c^	315.85 ± 18.26 ^b^	32.56 ± 0.93 ^c^	398.02 ± 8.24 ^b^	nd	5934.53 ± 182.59 ^a^	***
**Vanillic acid**	109.75 ± 1.22 ^b^	nd	nd	201.24 ± 2.64 ^a^	17.93 ± 1.44 ^c^	nd	185.74 ± 85.16 ^ab^	***
**Verbascoside**	nd	133.59 ± 3.04 ^d^	704.34 ± 39.10 ^c^	173.96 ± 4.40 ^d^	2535.70 ± 9.34 ^b^	2287.31 ± 29.72 ^b^	15,430.07 ± 3430.60 ^a^	***

^a–e^: Values with different letters are significantly different; significant levels: **: *p* < 0.01, ***: *p* < 0.001. ns: no significant. nd: not detected.

## Data Availability

The original contributions presented in the study are included in the article/Appendix A, further inquiries can be directed to the corresponding author.

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
