# Peer review of "Antioxidant Activity of Aqueous Extracts Obtained from By-Products of Grape, Olive, Tomato, Lemon, Red Pepper and Pomegranate"

_foods, 2024, doi:10.3390/foods13121802_

Round 1

Reviewer 1 Report

Comments and Suggestions for Authors

The antioxidant activity of extract from different by-products were compared in this manuscript. The idea is lack of novelty to some extent. So the authors should focus on the innovation information on their study including design and methods. Some points should be revised and improved as below.

1. Title, it is not accurate, please improve it. "by-products" should be moved before the fruits.

2. Abstract, line 9, what is "the one of rosemary extract"? 

3. introduction section, the authors made a detailed introduction one by one with the material by products, it is not essential, they should make a general description or introduction for industry.

4. Tables, they should be exihbited with 3-line table. And the significance label should be added.

5. Writing mistakes, line 141,142, ferric ion should be lower case for the valent. The authors should double-check others. And the reference list, the title should be consistent about the upper case or lower case, for example line 545. Too many references have been cited, please reduce them.

6. Conclustion can be compressed.

7. The language can be improved.

Author Response

Attached the reply to reviewer 1

Reviewer 2 Report

Comments and Suggestions for Authors

This manuscript compared the antioxidant activity of aqueous extracts obtained from grape, olive, tomato, lemon, red pepper and pomegranate by-products. This manuscript was written well. However, some details need to be added. Please consider the reviewer’s opinion in revising this manuscript to make it more comprehensive.

 General and specific comments:

1.     Section 1: This introduction does not show the need to do this work, please make changes. Why is it important to compare with rosemary extract?

2.     Please change the tables (1, 2, 3) to three-line table.

3.     The antioxidant capacity of the same substance is shown to be different. For example, LEM. it has high RSA, but lower MQA and RP. What is the reason?

Author Response

Attached the reply to reviewer 2

Reviewer 3 Report

Comments and Suggestions for Authors

The article foods-3014170 presented a study comparing the antioxidant activity and composition of aqueous extracts from residues/by-products of plant origin. The work is interesting, but some corrections are necessary.

Lines 20-22: Rephrase this sentence;

Lines 67-68: It is necessary to mention what the lemon residues are. Are these residues derived from the production of lemon juice? This is not clear in the text;

Lines 91-95: It is necessary to clarify further the novelty of this work in relation to others already published that evaluated aqueous extracts of plant origin;

Table 1: It is necessary to describe the methodologies used to determine the chemical composition of plant by-products;

LIne 165: Change in all text “M” to “mol/L”;

Table 2: it is necessary to make the sixth column wider to better highlight the negative values;

Table 4: It is necessary to insert the chemical structures of these compounds, even if only in the supplementary material;

It is necessary to describe in the text what characterizes these compounds as antioxidants. Although this is described in other works, it is worth mentioning in this article.

Conclusions: ok;

The number of references cited is very high.

Comments on the Quality of English Language

Minor editing of English language is required.

Author Response

Attached the reply to reviewer 3

Round 2

Reviewer 1 Report

Comments and Suggestions for Authors

The authors have made positivley  revision on the comments. Now it reads better. However, as the common research article, not review, the authors cited too many references. So if possible, reduce them, delete some less important or related literatures.

Author Response

The authors are very grateful for reviwer #1´s feedback. The authors have deleted some more cites at reference section.

Reviewer 2 Report

Comments and Suggestions for Authors

It could be published now.

Author Response

The authors are very grateful for reviwer #2´s feedback.

Reviewer 3 Report

Comments and Suggestions for Authors

The article is ok.

Author Response

The authors are very grateful for reviwer #3´s feedback.